# Deep Neural Network Architectures for Modulation Classification

**Xiaoyu Liu, Diyu Yang, Aly El Gamal**
School of Electrical and Computer Engineering
Purdue University
{liu1962, yang1467, elgamala}@purdue.edu

## Abstract

In this work, we investigate the value of employing statistical machine learning in general and deep learning in particular for the task of wireless signal modulation recognition. Recently in O'Shea & Corgan (2016), a framework has been introduced by generating a dataset using GNU radio that mimics the imperfections in a real wireless channel, and uses 10 different modulation types. Further, a convolutional neural network (CNN) architecture was developed and shown to deliver performance that exceeds that of expert-based approaches. We tested the architecture of O'Shea & Corgan (2016) and found it to achieve an accuracy of approximately 75% of correctly recognizing the modulation type. We find a design with four convolutional layers and two dense layers that gives an accuracy of approximately 83.8% at high SNR. We then develop architectures based on the recently introduced ideas of Residual Networks (ResNet, He et al. (2015)) and Densely Connected Networks (DenseNet, Huang et al. (2016)) and achieve high SNR accuracies of approximately 83.5% and 86.6%, respectively. We achieve the best accuracy of approximately 88.5% at high SNR by applying a Convolutional Long Short-term Deep Neural Network (CLDNN, Sainath et al. (2015)) to the modulation classification task. We then focus on the modulation types of QAM16 and QAM64 that were not well learned by neural networks and explore different statistical machine learning methods using expert features to classify them. We achieve an accuracy of 72 % in classifying QAM16 and QAM64 signals at high SNR using the combination of time and a high-order cumulant as expert feature.

## 1 Introduction

Signal modulation is an essential process in wireless communication systems. Modulation recognition tasks are generally used for both signal detection and demodulation. The signal transmission can be smoothly processed only when the signal receiver demodulates the signal correctly. However, with the fast development of wireless communication techniques and more high-end requirements, the number of modulation methods and parameters used in wireless communication systems is increasing rapidly. The problem of how to recognize modulation methods accurately is hence becoming more challenging.

Traditional modulation recognition methods usually require prior knowledge of signal and channel parameters, which can be inaccurate under mild circumstances and need to be delivered through a separate control channel. Hence, the need for autonomous modulation recognition arises in wireless systems, where modulation schemes are expected to change frequently as the environment changes. This leads to considering new modulation recognition methods using deep neural networks.

Deep Neural Networks (DNN) have played a significant role in the research domain of video, speech and image processing in the past few years. Recently the idea of deep learning has been introduced to the area of communications by applying convolutional neural networks (CNN) to the task of radio modulation recognition O'Shea & Corgan (2016). In this paper, we present our experiments of the deep neural network application on modulation recognition using optimized CNN, Densely connected network and CLDNN. We also explored a support vector machine method for recognizing QAM signals which were not well classified by neural networks.

## 2 SIMULATION SETUP

We use the RadioML2016.10b dataset generated in O'Shea & Corgan (2016) as the input data of our research. This dataset contains 10 types of modulations: eight digital and two analog modulations. These consist of BPSK, QPSK, 8PSK, QAM16, QAM64, BFSK, CPFSK, and PAM4 for digital modulations, and WB-FM, and AM-DSB for analog modulations. For digital modulations, the entire Gutenberg works of Shakespeare in ASCII is used, with whitening randomizers applied to ensure equiprobable symbols and bits. For analog modulations, a continuous voice signal is used as input data, which consists primarily of acoustic voice speech with some interludes and off times. The entire dataset is a 128-sample complex time-domain vector generated in GNU radio. 160,000 samples are segmented into training and testing datasets through 128-samples rectangular windowing processing, which is similar to the windowed continuous acoustic voice signal in voice recognition tasks. The training examples - each consisting of 128 samples - are fed into the neural network in 2*128 vectors with real and imaginary parts separated in complex time samples. The labels in input data include SNR ground truth and the modulation type. The SNR of samples is uniformly distributed from -20dB to +18dB. All training and testing are done in Keras using Nvidia M60 GPU. We use Adam Kingma & Ba (2014) from the deep learning library as optimizer in Keras and use Theano as back end.

## 3 RESULTS

We start with a basic two-convolutional-layer neural network, in which two convolutional layers with 256 1x3 filters and 80 2x3 filters, respectively, are followed by two dense layers. We then explore the effect of different filter settings by exchanging filter settings between the two convolutional layers. The performances of networks with different filter settings demonstrate that layer architectures with larger filters in earlier convolutional layers and smaller filters in deeper convolutional layers optimize the accuracy result at high SNR.

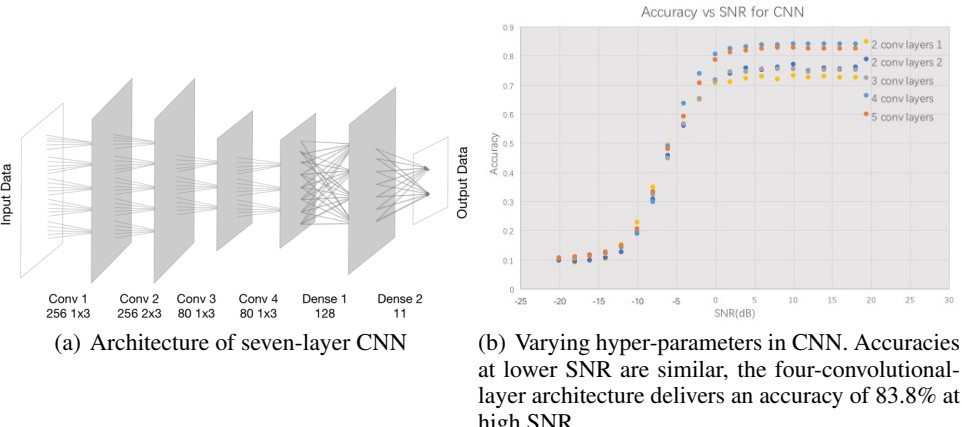

(a) Architecture of seven-layer CNN

(b) Varying hyper-parameters in CNN. Accuracies at lower SNR are similar, the four-convolutional-layer architecture delivers an accuracy of 83.8% at high SNR.

Figure 1: Architecture and performance of CNN

Next, we explore the optimal depth of CNN by increasing the number of convolutional layers from two to five. We find that the best accuracy at high SNR which is approximately 83.8% is obtained when using the four-convolutional-layer architecture as shown in Figure 1(a). This is a significant improvement of 8.8% over the two-convolutional-layer model. Due to the fact that lower loss corresponds to higher accuracy, a smoothly decreasing loss indicates that the network is learning well as it does for the four-convolutional-layer model. When the neural network gets deeper, it becomes less likely for the validation loss to converge. For the five and six-convolutional-layer models, large loss vibrations appear early during training, which means that the minimum losses achieved by these neural networks are larger than that of the four-convolutional-layer model, which leads to the poor classification performance.

We find that combining a residual network with the original CNN architecture demonstrates similar performance as the pure CNN architecture. Similar to the result of CNN, the best performance of 83.5% is achieved when we combine ResNet with a four convolutional layer neural network. Recognition accuracy also starts to decrease when we combine ResNet with a network architecture that has more than four convolutional layers.

Because more densely connected blocks require a deeper neural network, which in our experiments did result in accuracy degradation, we implement DenseNet on CNN architectures with only one densely connected block. We start with a three convolutional layer DenseNet and keep adding convolutional layers into the network until the accuracy result starts to descend. We achieve a best accuracy of 86.6% at high SNR using the four convolutional layer architecture shown in Figure 1(a).

We applied the CLDNN architecture and compared the performance of CLDNN with results demonstrated by ResNet and DenseNet. We added an LSTM unit into the network after the convolutional part. We believe that the cyclic connections extract relevant temporal features in the signal. The results of CLDNN - shown in Figure 2(a) - do outperform other models. The accuracy at high SNR reaches 88.5% and it is the highest among all tested neural network architectures. In Figure 2(b),

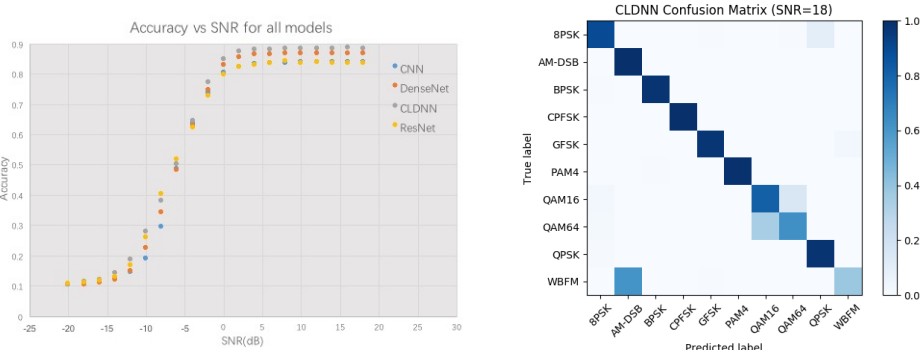

(a) Classification performance comparison between candidate architectures. CLDNN and DenseNet outperform other models with best accuracies of 88.5% and 86.6%, respectively.

(b) The confusion matrix of CLDNN at SNR=18 dB.

Figure 2: The performance of CLDNN

we show the classification results of the highest SNR case in a confusion matrix. There are two main discrepancies besides the clean diagonal in the matrix, which are WBFM being misclassified as AM-DSB and QAM16 being misclassified as QAM64. A small portion of 8PSK samples are misclassified as QPSK and a small portion of WBFM samples are misclassified as GFSK; we expect that further optimizing the neural network architecture and possibly increasing the depth would lead to capturing these subtle feature differences. We further notice that QAM16 and QAM64 are likely to be misclassified as each other, since their similarities in the constellation diagram make the differentiation vulnerable to small noise in the signal. We therefore explore different expert features to classify QAM signals.

We found that the popular cumulant feature used in previous work ( Marchand et al. (1998) Dobre et al. (2004) Aslam et al. (2012)) does not deliver good performance in the GNU radio generated data used in this work. We believe that this is mainly because in the theoretical models considered in the aforementioned works, the received signal is assumed to be stationary, which does not hold for real world data. As a result, we combine the time index with the cumulant as our new expert feature of each sample, and feed it into a support vector machine. Using this approach, we achieve a 72% classification accuracy result on QAM signals with SNR=18 dB.

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
