# OpenReview forum: "Deep Neural Network Architectures for Modulation Classification"
_ICLR.cc/2018/Workshop — Reject_

### Official Review · AnonReviewer1 · 2018-03-08
**comments**

**Rating:** 3
**Confidence:** 4

**Review:**

The paper reports modulation classification performance on RadioML2016 dataset using a variety of deep neural networks, including conventional CNN, ResNet, DCN and CLSTM.  Even though the results are good, it is hard to tell the novelty. The authors basically tried various network architectures and tuned configurations.  In addition, it was observed that a single model couldn't give the best performance across all the tasks (SNRs and types of modulation). So another classifier has to be trained to specifically classify the QAM signals.  It is not clear from the paper that how this is carried out. Is it a two-stage design, namely one first classifies the type of modulation and if it is QAM the signals go to the SVM with the expert features?
Overall, I find this work not very inspiring.

---

### Official Review · AnonReviewer2 · 2018-03-09

**Rating:** 3
**Confidence:** 3

**Review:**

This paper presents a deep neural network-based approach for frequency modulation classification. The proposed model is based on the CLDNN architecture used in ASR. The paper then presents a study on synthetic data.

This paper looks like a technical report, applying a well-known architecture (CLDNN) to an established task. Its novelty is thus very limited. Moreover, the experiments are not convincing. I therefore recommend to reject the paper.

Detailed review:

Section 2:
- The features used are not clear: is it FFT ? the complex spectrum seems to be used, the authors should explain why (my understanding is that keeping the signal phase is important for the task).

- The database seems to be split into train and test, without validation. This is a problem, as the architecture was tuned on the test set. Thus the main claim of the paper of reaching the best accuracy is questionable. The authors should use a validation set to tune the hyper-parameters and report the results on another test set.

Section 3 :
- the last paragraph is very hard to understand: the "cumulant" features should be presented, the classifier explained in more details and the experimental setup clarified (is it the same as the previous experiments?). The authors should also explain why they use a SVM in this case. The reported performance of 72% is not very informative by itself, it should be compared to the performance of the CLDNN model.

---

### Official Review · AnonReviewer3 · 2018-03-09
**A complete Deep Learning running on a complex task, but no proposition for new representation**

**Rating:** 3
**Confidence:** 3

**Review:**

This paper investigates deep learning for the task of wireless signal modulation recognition.
Several convolutional neural network (CNN) architecture was developed and shown to
deliver performance that exceeds that of expert-based approaches.
However the paper does not propose new representation nor training method.
It is then out of the topic of ICLR, maybe be better in a signal processing conference.

---

### Decision · Program_Chairs · 2018-03-20
**ICLR 2018 Workshop Acceptance Decision**

**Decision:**

Reject

**Comment:**

Based on the reviews, this paper has not been accepted for presentation at the ICLR workshop. However, the conversation and updates can continue to appear here on OpenReview.